# Polysulfone–Polyvinyl Pyrrolidone Blend Polymer Composite Membranes for Batik Industrial Wastewater Treatment

**DOI:** 10.3390/membranes11010066

**Published:** 2021-01-18

**Authors:** Arifina Febriasari, Annisa Hasna Ananto, Meri Suhartini, Sutrasno Kartohardjono

**Affiliations:** 1Chemical Engineering Department, Faculty of Engineering, Universitas Indonesia, Kampus UI, Depok 16424, Indonesia; arifina.febriasari@ui.ac.id (A.F.); huriya@ui.ac.id (H.); annisa.hasna@ui.ac.id (A.H.A.); 2National Nuclear Energy Agency, Jl. Lebak Bulus Raya No. 49, Jakarta 12440, Indonesia; meri@batan.go.id

**Keywords:** batik wastewater treatment, blend polymer composite, polysulfone membrane, PVP, antifouling performance

## Abstract

Batik wastewater, in general, is colored and has high concentrations of BOD (biological oxygen demand), COD (chemical oxygen demand), and dissolved and suspended solids. Polysulfone (PSf)-based membranes with the addition of polyvinyl pyrrolidone (PVP) were prepared to treat batik industrial wastewater. PSf/PVP membranes were prepared using the phase inversion method with N-methyl-2 pyrrolidone (NMP) as the solvent. Based on the membrane characterization through FESEM, water contact angle, porosity, and mechanical tests showed a phenomenon where the addition of PVP provided thermodynamic and kinetic effects on membrane formation, thereby affecting porosity, thickness, and hydrophilicity of the membranes. The study aims to observe the effect of adding PVP on polysulfone membrane permeability and antifouling performance on a laboratory scale through the ultrafiltration (UF) process. With the addition of PVP, the operational pressure of the polysulfone membrane was reduced compared to that without PVP. Based on the membrane filtration results, the highest removal efficiencies of COD, TDS (total dissolved solid), and conductivity achieved in the study were 80.4, 84.6, and 83.6%, respectively, on the PSf/PVP 0.35 membrane operated at 4 bar. Moreover, the highest color removal efficiency was 85.73% on the PSf/PVP 0.25 operated at 5 bar. The antifouling performance was identified by calculating the value of total, reversible, and irreversible membrane fouling, wherein in this study, the membrane with the best antifouling performance was PSf/PVP 0.25.

## 1. Introduction

Batik wastewater is usually obtained from the remaining water from the dyeing and washing process. This wastewater contains chemicals added during the process, such as dyes, starch, oil, wax, lye, and detergents [1]. These substances are generally non-biodegradable, so that they will accumulate in soil and water and cause environmental pollution. Textile industrial wastewaters, in general, are colored and have high levels of biological oxygen demand (BOD), chemical oxygen demand (COD), and dissolved and suspended solids. Dipping wastewater has a characteristic temperature of 29–35 °C, pH ranges from 6.8 to 10.5, COD of 1712.7–1793 mg/L, BOD of 159.7–168 mg/L, and TSS (total suspended solid) of 1233.7–1317 mg/L [2]. Separation technologies that are currently studied by researchers for processing industrial textile wastewater include coagulation/flocculation, ozonation, and membrane technology [3,4,5]. Several researchers have examined the type of membrane technology adequate for processing industrial textile wastewater, including ultrafiltration, nanofiltration, reverse osmosis, membrane distillation, and the use of a membrane bioreactor [6,7,8,9,10]. Ultrafiltration membranes are generally used for separating suspended solids and are particularly suitable for reducing the concentration, purification, and fractionation of macromolecules such as proteins, substances, colors, and other polymeric materials. Dyes for coloring has particles with sizes of 0.5 to 2.0 microns, so they correspond to the pore sizes of the ultrafiltration membrane, which are 0.001–0.05 μm [11]. 

Membrane filtration processes are often chosen for economic and environmental reasons. However, the membrane has a disadvantage in terms of the limited time that it can be used due to membrane fouling [12,13,14,15]. Nanofiltration and reverse osmosis membranes can remove ions, salinity, and macromolecules in water but are very susceptible to fouling. Therefore, a pre-treatment process is accomplished in the form of ultrafiltration, which can remove impurities that cause fouling on the nanofiltration membrane, and reverse osmosis, which removes suspended solids, colloids, and high molecular mass organic materials [16]. Another combined process was also adopted to minimize this fouling problem in textile wastewater treatment, including combining coagulation and membrane filtration [17,18,19].

Membrane material is an essential factor, considering chemical, thermal, and mechanical stability. Polysulfone is one of the materials commonly used for membrane filtration. Polysulfone has good resistance to alkaline solutions; thus, it is suitable for batik waste [20]. Therefore, membrane preparation is achieved using polysulfone polymers, which have characteristics of excellent stability, wide pH range (pH 2–13), resistance to oxidants, and ease of modification, so that they are often used in the manufacture of membranes [21]. However, polysulfone membranes have low antifouling performance and low membrane flux values due to their low hydrophilic nature [22]. The addition of hydrophilic polymers to the printing solution is often performed to increase membrane permeability and antifouling ability. Commonly used additives include polyvinyl pyrrolidone (PVP) and polyethylene glycol (PEG). Many studies have shown that the addition of PVP between 3 and 90 wt.% of the total doped solution can increase the hydrophilicity, permeability, and antifouling performance of polysulfone membranes [23,24]. The study aims to observe the effect of adding PVP on the permeability and antifouling performance of the PSf membrane on a laboratory scale through the ultrafiltration (UF) process. Before the UF process, the batik wastewater was pretreated through the coagulation and flocculation process using polyaluminum chloride (PAC) coagulants. PAC was chosen as it has many advantages such as fewer aluminum residuals, better performance at low temperature, less sludge volume, less effect on raw water’s pH value, and more rapid flocculation [25].

## 2. Materials and Methods 

### 2.1. Materials

Polysulfone (PSf, UDEL P-1700 NT11) was provided from Solvay (Brussels, Bel-gium). Polyvinyl pyrrolidone (PVP, MW: 29,000 g/mole), as an additive, was provided from Sigma Aldrich (St. Louis, MI, USA), and N-methyl-2 pyrrolidone (NMP), as a solvent, was purchased from Merck Indonesia. COD reactor digestion vials (Low range, pk/150, method 8000) were provided from HACH, USA. Pt/Co color standard was also purchased from Merck (Darmstadt, Germany).

### 2.2. Batik Wastewater Pre-Treatment

Batik wastewater pre-treatment was conducted by the coagulation–flocculation method. To carry out this method, 300 mL of batik wastewater was put into a beaker at room temperature, and then the pH of the wastewater was adjusted to pH 4. Polyaluminum chloride (PAC) was added in the amount of 500 ppm of the waste solution and then stirred at 200 rpm for 1 h. The stirring speed was then decreased to 40 rpm for 10 min. The solution was allowed to stand for 30 min for the settling process. The precipitated solution was then filtered with qualitative filter paper to separate out the suspended solids. Resulting pre-treatment batik wastewater samples were analyzed for pH, COD, TSS, color, and TDS concentrations using the methods listed in Table 1.

### 2.3. Membrane Preparation

Membrane preparation was performed by the phase inversion method [14]. The doped solution was made by slowly adding PVP and polysulfone polymers to the NMP solvent in the ratios listed to Table 2, while stirring at 200 ppm for 30 min. After 30 min, the stirring speed was increased to 300 rpm, and the temperature was raised from room temperature to 60 °C. The temperature was maintained at 60 °C until the solution was homogeneous. After the solution’s viscosity increased, the stirring speed was increased to 500 rpm for 5 h. The doped solution was then allowed to stand at room temperature for 1 h to remove air bubbles that had formed. The doped solution was poured onto a glass plate, flattened with a glass rod to form a thin layer, and then let to stand for 8 min. The liquid layer on the glass plate was soaked in a coagulation bath filled with pure water until a membrane sheet was formed, and then it was allowed to stand in the coagulation bath overnight. The membrane sheet was then immersed in 50% ethanol solution for 1 h and immersed again in 96% ethanol for 30 min. The membrane was then dried at room temperature. An illustration of the preparation process is shown in Figure 1. Ethanol (50%) was used to remove residual organic solvent from the surface and the pores of the membranes. A higher concentration of ethanol (96%) was used at the last stage to ensure that no organic solvent was left on the membrane surface or in the pores and to speed up the drying process of the membrane at room temperature. The membrane preparation was repeated three times for each membrane variant.

### 2.4. Membrane Characterization

Membrane characterization was performed to identify morphology, functional group changes, hydrophilicity, and membrane porosity. Membrane morphology was investigated by field emission scanning electron micrograph (FESEM, FEI inspect F50). Sample specimens were thinly coated with Au and observed at an excitation voltage of 20 kV. Changes in membrane functional groups were observed with attenuated total reflectance Fourier transform infrared (ATR-FTIR) spectra. The spectra were accomplished using the Nicolet ™ iS50 FTIR Spectrometer with the NIR module and a wavenumber range of 600–4000 cm^−1^.

Moreover, to observe membrane hydrophilicity, a water contact angle test was performed. This test was accomplished by dripping deionized water on the surface of the membrane. The contact angle was recorded three times randomly and measured with an angle meter. To measure the overall porosity of the membrane, the wet–dry weight method was used. The overall porosity of the membrane was calculated using the following equation [26]:(1)$$\varepsilon \;\left(\%\right)=\frac{W_{w}-W_{d}}{\rho_{w}\cdot A\cdot l}\times 100$$
where *W_w_* is the weight of the wet membranes that were boiled in deionized water for 2 h and then allowed to soak in water for 24 h, *W_d_* is the dry membrane weight, *ρ**_w_* is the density of deionized water, *A* is the membrane area, and *l* is the membrane thickness. Two comparable methods measured membrane thickness, i.e., measurement on SEM results and measurement using Dial Thickness (Pressier 307822).

The average membrane pore size was measured using the Guerout–Elford–Ferry formula through the filtration velocity method. The calculation of the membrane pore size average (*r_m_*) was obtained by the following equation [27].
(2)$$r_{m}=\sqrt{\frac{\left(2.9-1.75\varepsilon \right)\times 8\eta lQ}{\varepsilon \times A\times \Delta P}}$$
where ε is the porosity, η is the viscosity of pure water (8.9 × 10^−4^ Pa.s), *l* is the thickness of the membrane, *Q* is the volume of permeate per unit time (m^3^.s^−1^). *A* is the effective surface area of the membrane, and Δ*P* is trans membrane pressure (700 kPa).

The membrane tensile strength was observed in this study to determine the effect of PVP on the mechanical strength of the PSf membrane. The test was carried out using a Universal Tensile Meter (Shimadzu 10 kN) with the standard method of ASTM D882-12. The conditions were performed at room temperature with a 50 mm/min test speed and a distance between the grips of 100 mm. The membrane tensile strength value was calculated by the following equation [28]:(3)$$\sigma =\;\frac{F}{A_{0}}$$
where *σ* is the tensile strength (MPa), *F* is the load applied during measurement (N), and *A*_0_ is the area of the membrane specimen (m^2^). The percentage of elongation at break was obtained from the ratio between the initial and extended length at the time of breakage of the tested specimen at room temperature, and was calculated using the following [29]:(4)$$\epsilon =\;\left(\frac{\Delta L}{L_{0}}\right)\times 100$$
where ε is the percentage of elongation at break (%), Δ*L* is the extended length of the membrane during measurement (mm), and *L*_0_ is the length of the original membrane measured according to the distance between the grips (mm).

### 2.5. Pure Water and Permeate Flux

The membrane performance test was conducted by a cross-flow ultrafiltration system using a pump with variations in feed pressure and time. The membrane, with an effective area of 10.2 cm^2^, was placed in a membrane cell permeation system, which included a wastewater tank in the feed, a permeate tank, and a retentate tank to accommodate wastewater that did not penetrate the membrane. The operating pressure range of the driving force used was 4–7 bar. The pressure was used for pure water flux, batik wastewater flux (permeate flux), and water flux after backwashing. The pressure was applied except for with the permeate flux on the pristine PSf membrane, where the minimum pressure that could be used to penetrate the membrane was 12 bar. Batik wastewater after pre-treatment was used for membrane performance testing. The permeate flux of the experiment was calculated using the following equation [30]:(5)$$J=\;\frac{V}{A\cdot t}$$
where *J* is the flux in mL.cm^−2^s^−1^, *V* is the volume of the permeate penetrating through the membrane (mL), *A* is the effective area of the membrane (cm^2^), and *t* is time (s). 

### 2.6. Batik Wastewater Removal

The removal efficiency was calculated by the following [31]:(6)$$R\;\left(\%\right)=\;\left(1-\frac{C_{p}}{C_{f}}\right)\times 100$$
where *R* is the percent waste removal (COD, TDS, conductivity, and color), *Cp* is the concentration of substances in the permeate phase, and *Cf* is the concentration of substances in the feed phase. COD and color were measured by the UV–Vis spectrophotometric method (Thermo fisher scientific, UV23000) using reactor digestion vials (method 8000). Moreover, TDS, conductivity, and pH were measured using a water quality meter (AZ 86031).

### 2.7. Membrane Fouling Study

The ability of membranes to overcome fouling problems was compared by calculating membrane fouling values. After the wastewater filtration process, the membrane was backwashed with deionized water in 4 bar for 30 min. Membrane fouling was then obtained from the values of pure water flux, which started to become fouled (*J_W_*_1_), permeate flux value (*Jp*), and flux value after being backwashed (*J_W_*_2_), which were calculated by the following [32]:(7)$$R_{r}=\;\left(\frac{J_{W2}-J_{P}}{J_{W1}}\right)\times 100$$
(8)$$R_{ir}=\left(\frac{J_{W1}-J_{W2}}{J_{W1}}\right)\times 100$$
(9)$$R_{t}=\left(\frac{J_{W1}-J_{P}}{J_{W1}}\right)\times 100\%$$
where *R_r_* is reversible membrane fouling, *R_ir_* is irreversible membrane fouling, and *R_t_* is total membrane fouling.

## 3. Results and Discussion

### 3.1. Membrane Characterization

Membrane characterization by FESEM is given in Figure 2 and Figure 3. This study’s preparation resulted in ultrafiltration membrane with an asymmetric structure with non-uniform pores distributed on the top surface and sponge-like macro voids in the sublayer. As shown in Figure 2 and Figure 3, the changes in the membrane after the addition of PVP could be seen in the formation of pores in the image with 2500 times magnification. The addition of 0.25 g PVP to the PSf membrane increased the pore structures in the membrane’s top surfaces [33]. PSf/PVP 0.35 membrane, although it appears to have a larger average pore size compared to pristine PSf (Figure 3 and Table 3), the addition of a larger PVP concentration has an effect of increasing thickness, so that porosity decreases (Table 3) [34].

The pristine PSf membrane had a high surface tension against organic solvents (41 mN/m) and aromatic groups, making it quicker to settle and tighten the molecules during the coagulation process [35,36]. The addition of PVP to the doped solution has a two-fold effect. Thermodynamically, due to the hydrophilic nature of PVP, it facilitated penetration into the print and sped up solvent and non-solvent exchange. However, kinetically, using PVP can increase the viscosity of the solution [37]. On the PSf/PVP 0.25 membranes, the thermodynamic effect of PVP was more dominant, so that the de-mixing process suddenly occurred; thus, the pores were more formed. In fact, on the PSf/PVP 0.35 membrane, the kinetic effect of high solution viscosity overcame the thermodynamic effect of PVP to inhibit non-solvent diffusion, and a thicker layer membrane was formed. This was confirmed in the analysis of the membrane’s overall porosity, as seen in Table 2, where the PSf/PVP 0.25 membrane had a higher porosity than the PSf/PVP 0.35 membrane. An illustration of the differences in the PSf membrane before and after PVP addition can be seen in Figure 2.

PVP is water-soluble, which must be considered in the formation of pores in the membrane surface. During the membrane coagulation process in deionized water, solvent exchange occurs, so that some of the PVP dissolves in the water and leaves PSf. This phenomenon is predicted to increase the formation of non-uniform pores on the top surface of the PSf/PVP membrane (Figure 2 and Figure 3) [38].

The difference in thickness values obtained from measurement of FESEM images and measurements using dial thickness indicates that the membranes have a non-uniform thickness. However, the change in thickness and porosity values between the two measurements shows the same trend. Based on the results shown in Table 3, the difference in membrane thicknesses of pristine PSf and PSf/PVP 0.15 was not significant. However, the overall porosity and average pores size increased in the PSf/PVP blend membrane compared to the PSf pristine (Table 3). On the PSf/PVP 0.15 membrane, it was seen that the average pore size was reduced compared to the PSf/PVP 0.15 membrane (17 nm to 13 nm), but the porosity and permeability values increased. This is also indicated by an increase in pure water flux on the PSf/PVP 0.25 membrane compared to PSf/PVP 0.15 by three times (from 2.9 × 10^−3^ to 6.1 × 10^−3^ cm^3^/cm^2^ s). 

The membrane’s hydrophilicity was identified based on the water contact angle (WCA) measurements shown in Table 3. Pristine PSf membranes showed hydrophobic properties with a WCA value greater than 90°. The addition of PVP caused the membrane surface to become more hydrophilic, as indicated by a decrease in the WCA value. The more PVP that was added, the more it showed a tendency to have hydrophilic properties on the membrane surface [39,40,41]. The decrease in WCA from PSf/PVP 0.25 to PSf/PVP 0.35 was very low. WCA values were measured on the membrane surface. In the PSf/PVP 0.35 membrane, where the kinetic effect of PVP was higher and produced a thicker membrane, the PVP distribution was not only on the membrane surface. This caused the WCA value detected on the surface of the PSf/PVP 0.35 membrane to be almost the same as that on the PSf/PVP 0.25 membrane.

A comparison of the results of ATR-FTIR spectra is presented in Figure 4. The carbonyl bond peak in pure PVP appeared quite clearly at the wave number 1639 cm^−1^ [24]. This peak also appeared and increased in intensity on the PSf membrane with the addition of PVP (0.25 and 0.35 g) compared to the pristine PSf membrane, although it was not significant. Moreover, the peaks at wave numbers 1578 cm^−1^ (C=C aromatic vibration) and 1228 cm^−1^ (C–O–C bonds) were the specific peaks of the PSf membrane [24,42]. The very weak peak of the PSf/PVP membranes in the wavenumber of 1639 cm^−1^ was due to the low concentration of PVP added to the PSf membrane. There is a possibility that this phenomenon was also due to the nature of PVP, which has a high solubility in water. Thus, some of the PVP was rinsed during the immersion process with water and ethanol. However, considering the results of hydrophilicity characterization (WCA in Table 3) and an increase in membrane flux, this proves that some PVP still existed in the blend membrane, and it was predicted that part of the PVP was capsulated by PSf on the membrane surface [38,43,44].

Table 4 shows the mechanical strength of the membranes. It can be seen that the addition of PVP reduced the tensile strength of the membrane, as predicted. This occurred due to the increase of pores formed in the membranes with PVP addition [45]. Elastic deformation of the membrane decreased with the addition of PVP; hence, the membrane was more liable to swelling. Moreover, the percentage value of elongation at the break was not the same as the value of tensile strength on the membrane. On the PSf/PVP 0.15 membrane, percent elongation at break increased compared to the pristine PSf membrane, although the tensile strength value decreased. The phenomena appeared due to the increase of the membrane’s plasticity with the addition of PVP at a certain level [24]. A decrease in elongation at break in the PSf/PVP 0.25 membrane and an increase in the PSf/PVP 0.35 membrane identified that the percent of elongation at break could be influenced by the viscosity character of the membrane doped solution [46]. In line with the discussion on FESEM characterization results, the addition of PVP to the PSf/PVP 0.35 membrane had a more dominant kinetic factor than the thermodynamic factor.

### 3.2. Batik Wastewater Pre-Treatment

The specific test of the initial Batik wastewater was conducted to obtain an initial reference to measure the effectiveness of the waste treatment process. The characteristics tested were COD, TDS, TSS, conductivity, color, and pH. The treated wastewater was from the leaching process after the coloring process in the batik factory. Before being processed, batik wastewater was filtered first and then diluted one- to fourfold. Dilution is performed as a primary waste treatment to minimize the effects of waste pollution and reduce the concentration when going through the coagulation–flocculation process. If the waste has a high concentration, a hefty dose of coagulant is needed for efficient floc formation [47].

The pre-treatment process was conducted by coagulation–flocculation in the jar test using PAC as a coagulant. PAC is a type of coagulant derived from aluminum salts, often used for textile industrial wastewater treatment as a pre-hydrolyzed coagulant. PAC has advantages such as low alkalinity consumption and produces less sludge than non-hydrolyzed coagulants [48,49]. Wastewater was conditioned based on an optimum pH of 4 and an optimum coagulant dose of 500 ppm [50]. Afterward, the wastewater was filtered with rags. Pre-treatment was performed to remove floc, which could damage the membrane and cause fouling. The pH of the pre-treated waste was changed to pH 7 so as not to damage the membrane. This pH change caused sediment to appear so that the waste was filtered again using filter paper. The filtrate water then became the feed for the membrane filtration process. The characterization of batik waste is presented in Table 5. Pre-treatment successfully reduced COD, TSS, color, and conductivity, but increased TDS. The increase in TDS after pre-treatment was caused by the reduction of pH to pH 4, which caused salt formation due to the addition of acid to the waste, which was initially alkaline [50]. The same phenomena also occurred when setting it to pH 7; salt formation from the neutralization process caused flocs, which increased the TDS, TSS, conductivity, and color.

### 3.3. Pure Water Flux

The pure water fluxes conducted at various feed pressure variations of 4, 5, 6, and 7 bar in a cross-flow system characterized the membrane performance test. The pure water flux profiles with time at various membranes and feed pressures are shown in Figure 5 and Figure 6, respectively. The pure water flux profiles in Figure 5 and Figure 6 confirm the membrane’s porosity, where the higher the porosity of the membrane, the higher the value of pure water flux [51,52,53]. The PSf/PVP 0.25 membrane was confirmed to have the highest pure water flux due to the highest porosity. The increase in the feed pressure from 4 to 5 bar had an insignificant effect on the pure water flux. However, after increasing the feed pressure to 6 and 7 bar, the effect was more significant. 

### 3.4. Batik Wastewater Flux

One of this study’s objectives was to see the effectiveness of adding PVP to wastewater treatment applications, especially in the batik industry, at lower operational pressures. This batik wastewater flux study proved that the addition of PVP to polysulfone membranes can be operated at lower pressures than polysulfone membranes without PVP. It could provide preliminary evidence that the application of PSf/PVP membranes for wastewater treatment could provide safe energy [54]. Simultaneously, the PSf/PVP membranes were applied at a pressure of 4–7 bar with a permeate flux range from 9.8 × 10^−5^ to 16.7 × 10^−4^ cm^3^·cm^−2^·s^−1^. The permeate flux on the PSf/PVP 0.35 membrane was lower, with a significant difference in reduction compared to the PSf/PVP 0.15 and PSf/PVP 0.25 membranes, as presented in Figure 7. Based on the batik wastewater flux profile presented in Figure 8, the pristine PSf membrane cannot operate at pressures below 12 bar with a permeate flux range from 1.8 × 10^−5^ to 3.34 × 10^−5^ cm^3^·cm^−2^·s^−1^.

### 3.5. Pollutant Removal

The membrane’s ability to remove pollutants in batik wastewater in various feed pressures is demonstrated in Figure 9. Feed pressure of 4 to 7 bar was applied to PSf/PVP membranes, while feed pressure of 12 bar was applied to pristine PSf membrane. Figure 9 describes the membrane behavior of each waste parameter. In COD and TDS removal, the PSf/PVP 0.35 membrane had the highest removal efficiency. It indicated that the pore size affected the removal process, where it was previously confirmed that the PSf/PVP 0.35 membrane had the lowest porosity compared to other membranes [55]. Moreover, the highest color and conductivity removal efficiency were given by the PSf/PVP 0.25 membrane. The process of removing color and conductivity depends on the charge of the membrane surface [56]. Based on a study by Kajekar et al. (2015), the surface of the PSf/PVP blend polymer composite membrane at a pH above 5 has a negative zeta potential value, and this indicates that there is an electrostatic potential that plays a role when absorbing dyes on the membrane surface [57]. In this case, with a particular composition, the PSf/PVP membrane had a more capable electrostatic contribution, removing the dye and reducing wastewater conductivity.

Water samples of fresh batik wastewater after pretreatment and after membrane filtration processes operating at 12 bar for pristine PSf and various PSf/PVP membranes operating at 5 bar are presented in Figure 10. Figure 10 shows the difference in the clarity of fresh wastewater with other water samples. However, it is not easy to distinguish the clarity between water after pretreatment and water permeate after membrane filtration in plain view.

### 3.6. Membrane Resistance

The results of total membrane fouling (*R_t_*), reversible membrane fouling (*R_r_*), and irreversible membrane fouling (*R_ir_*) are shown in Figure 11. The high value of total and irreversible membrane fouling indicates low antifouling membrane performance. High reversible membrane fouling identifies that the membrane has a good antifouling performance [32]. Based on the calculation results of membrane fouling in Figure 9, it can be concluded that the PSf/PVP 0.25 membrane had the best antifouling performance. Moreover, the highest irreversible resistance was in the pristine PSf membrane.

Figure 12 illustrates how the membrane can leave a fouling component that cannot be removed by the backwash process known as irreversible fouling. The greater the potential to leave irreversible fouling, the lower the antifouling performance of the membrane [58]. Moreover, the fouling component that can be removed after the backwash process is reversible fouling. Figure 13 shows the conditions of the pristine PSf and PSf/PVP membranes after the filtration process and after backwash. After the filtration process in the pristine PSf membrane, some pores were seen to be blocked by the fouling component. After the backwash process, a few pores were open, indicating that the reversible fouling was removed. The line marks in the pristine PSf membrane after backwash occurred due to the high pressure applied during the filtration and backwash processes, whereas FESEM of the PSf/PVP membrane (after backwash) showed that the fouling component left on the membrane surface was smaller compared to the pristine PSf. The higher hydrophilicity and lower operating pressure in the cross-flow system minimized the fouling component being forced into the membrane pores, which can cause irreversible fouling [59,60].

## 4. Conclusions

The experiments were conducted to treat wastewater from the batik industry through a polysulfone-based membrane separation process. This study presents research results proving that the addition of a very low amount of PVP (less than 1 wt.% of the total weight of the doped solution) can still affect the increase in membrane flux and hydrophilicity values. This also affected the decrease in operating pressure required for filtration of batik wastewater after pre-treatment. The results also confirmed that polysulfone-based membranes can be modified by adding PVP by the phase inversion method. The membrane characterization results verified that the addition of PVP has two roles, both from thermodynamic and kinetic aspects. In terms of thermodynamics, PVP increased pore formation and membrane hydrophilicity, although the membrane’s thickness and tensile strength were reduced. Kinetically, it increased the viscosity of the membrane doped solution; hence, the thickness and plasticity of the membrane increased. Both results were shown by FESEM analysis, porosity, water contact angle, and tensile strength test. The solubility of PVP in water is also considered a cause of pore formation in membrane surfaces, and the phenomenon was confirmed by FTIR analysis. The addition of PVP reduced the membrane’s operating pressure by 67%, with COD, color, TDS, and conductivity removal efficiency of 80.4, 85.7, 84.6, and 83.6%, respectively. The antifouling performance was studied, and it was found that the membrane with the highest antifouling performance was PSf/PVP 0.25, with a reversible membrane fouling of 74.04%.

## Figures and Tables

**Figure 1 membranes-11-00066-f001:**
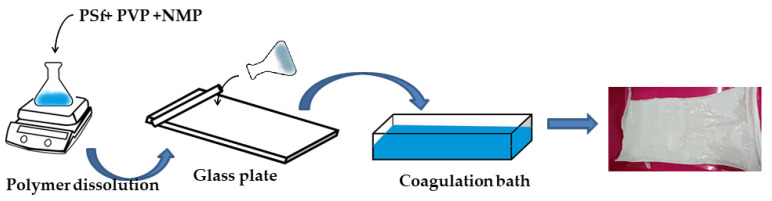
Illustration of membrane preparation.

**Figure 2 membranes-11-00066-f002:**
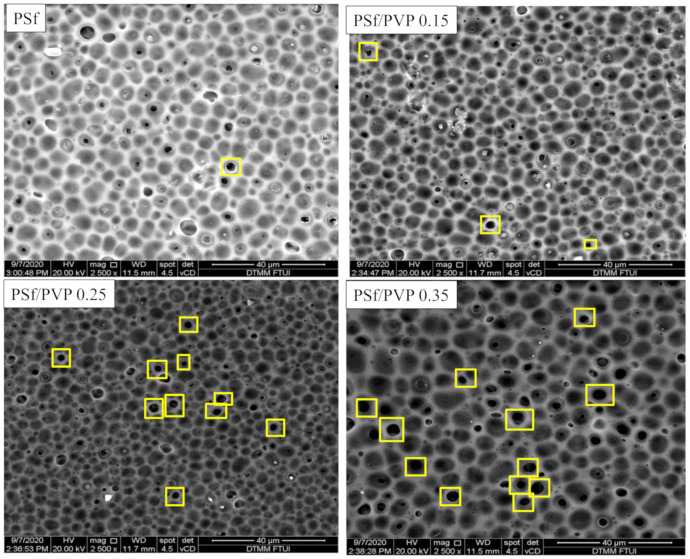
FESEM characterization of the surfaces of PSf, PSf/PVP 0.15, PSf/PVP 0.25, and PSf/PVP 0.35 membranes with a magnification of 2500 times.

**Figure 3 membranes-11-00066-f003:**
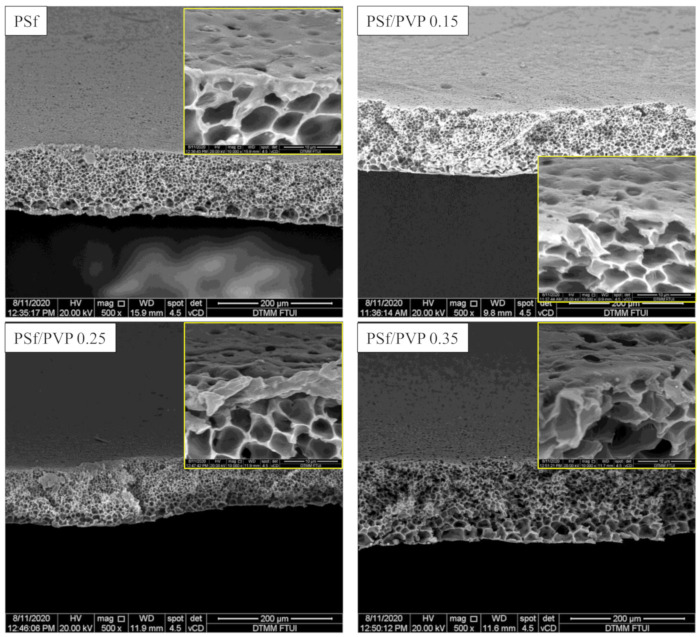
FESEM characterization of the cross sections of PSf, PSf/PVP 0.15, PSf/PVP 0.25, and PSf/PVP 0.35 membranes with a magnification of 500 times.

**Figure 4 membranes-11-00066-f004:**
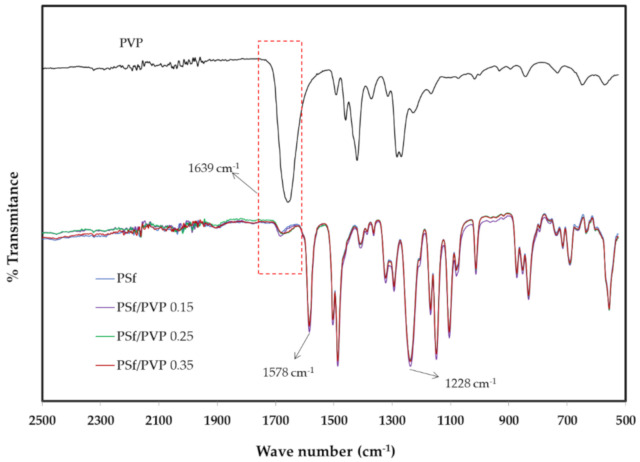
Attenuated total reflectance Fourier transform infrared (ATR-FTIR) spectra comparison of the membranes.

**Figure 5 membranes-11-00066-f005:**
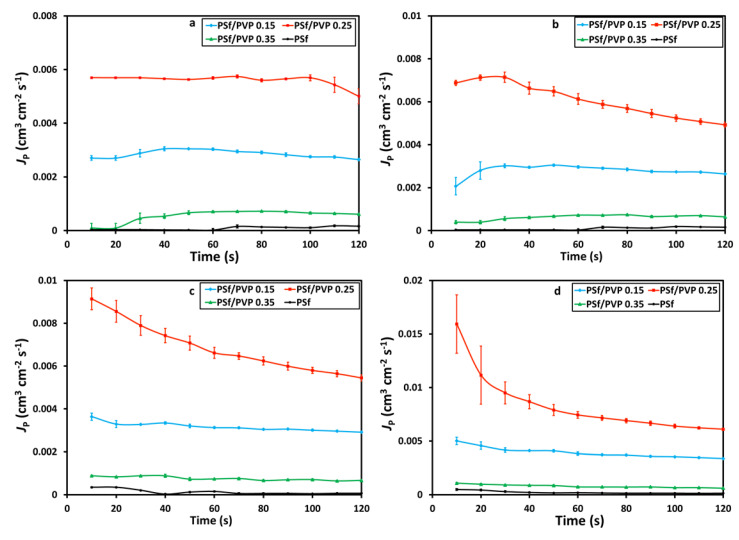
The profile of pure water flux (cm^3^·cm^−2^·s^−1^), *J*_P_, with time at the feed pressure of (**a**) 4 bar, (**b**) 5 bar, (**c**) 6 bar, and (**d**) 7 bar on the membrane filtration process.

**Figure 6 membranes-11-00066-f006:**
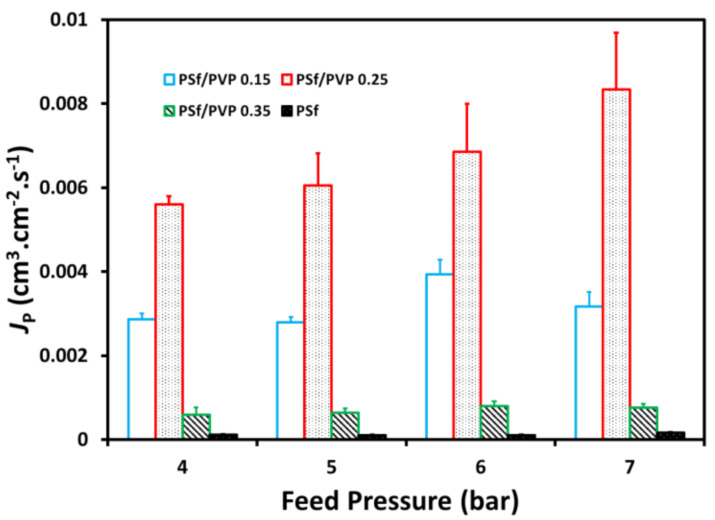
Effect of feed pressure on pure water flux, *J*_p_.

**Figure 7 membranes-11-00066-f007:**
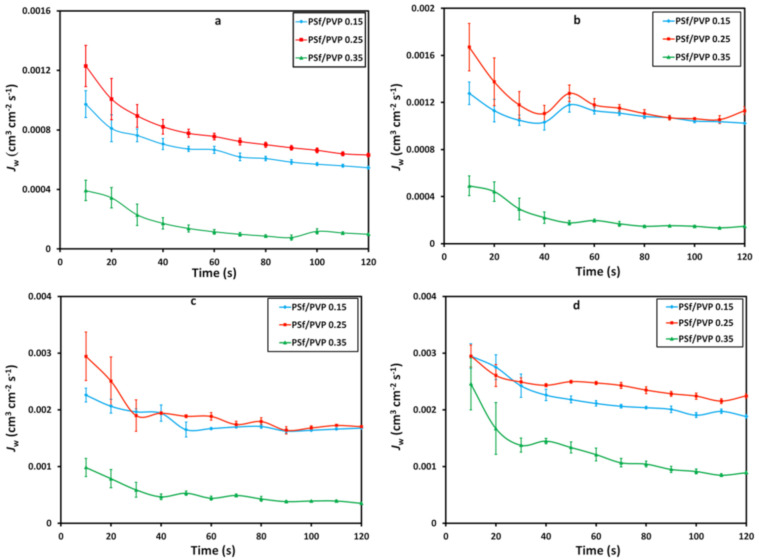
The profile of wastewater flux (cm^3^·cm^−2^·s^−1^), *J*_W_, with time at feed pressure of (**a**) 4 bar, (**b**) 5 bar, (**c**) 6 bar, and (**d**) 7 bar on the membrane filtration process.

**Figure 8 membranes-11-00066-f008:**
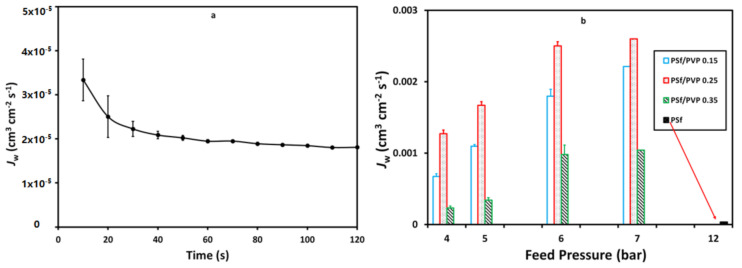
Profile of wastewater flux, *J*_W_, of the pristine PSf membrane at 12 bar (**a**) and the effects of feed pressure on the wastewater flux (**b**) on the membrane filtration process.

**Figure 9 membranes-11-00066-f009:**
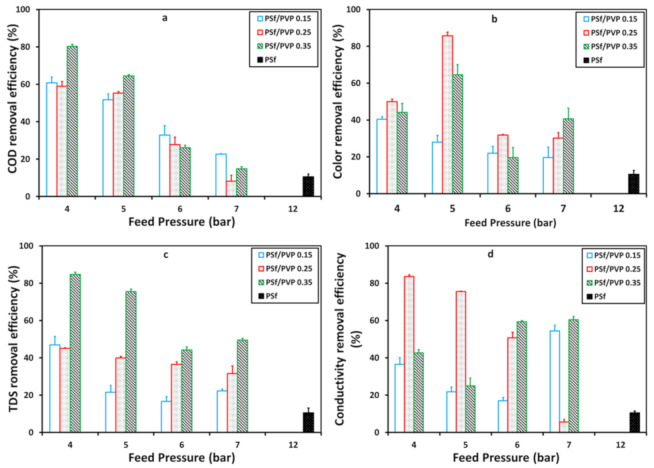
The membrane removal efficiency profile of chemical oxygen demand (COD) (**a**), color (**b**), total dissolved solid (TDS) (**c**), and conductivity (**d**) with feed pressure on the membrane filtration process.

**Figure 10 membranes-11-00066-f010:**
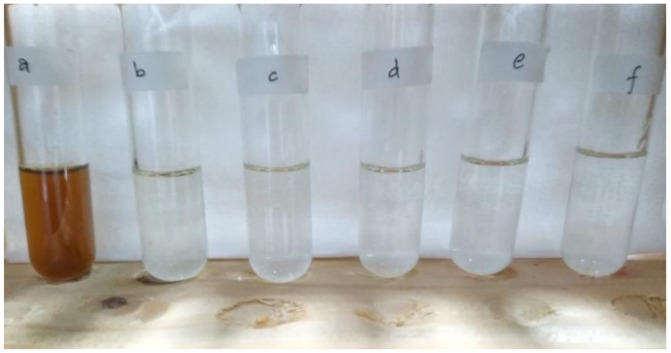
Water samples taken from (**a**) fresh batik wastewater and (**b**) after pretreatment, and permeated feed samples from different membranes: (**c**) pristine PSf operating at a pressure of 12 bar, (**d**) PSf/PVP 0.15, (**e**) PSf/PVP 0.25, (**f**) PSf/PVP 0.35 operating at 5 bar, respectively.

**Figure 11 membranes-11-00066-f011:**
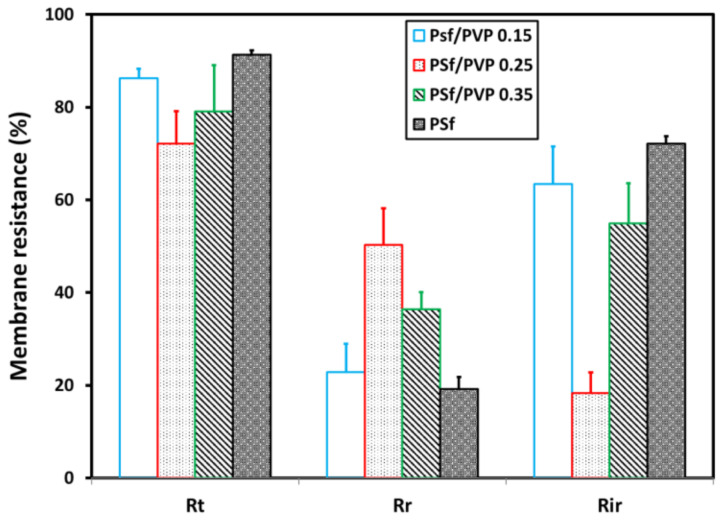
Membrane fouling comparison of various membranes.

**Figure 12 membranes-11-00066-f012:**
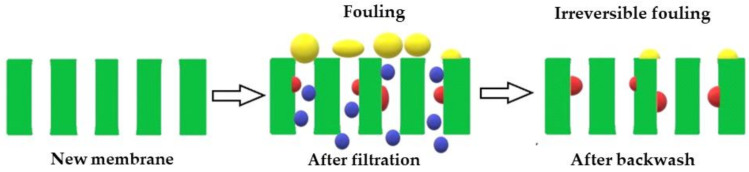
Illustration of membrane fouling and irreversible fouling after filtration and after backwash.

**Figure 13 membranes-11-00066-f013:**
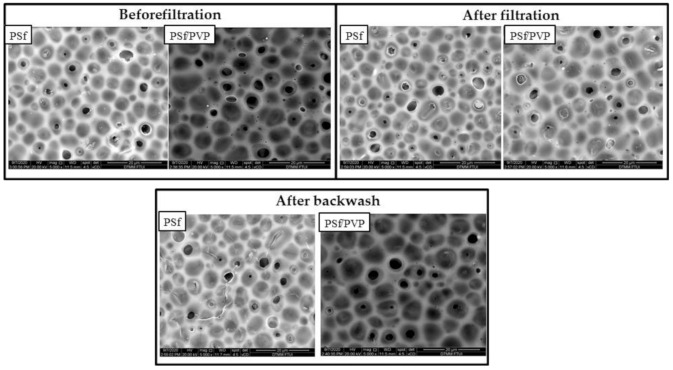
Comparison of FESEM on the pristine PSf and PSf/PVP blend polymer composite membrane before filtration, after filtration, and after backwash.

**Table 1 membranes-11-00066-t001:** Analytical methods and equipment used for water quality indicator measurements.

Indicator	Analysis Method	Equipment
pH	Electrometric	Water quality meter (AZ 86031)
COD	Digestion reaction and UV visible	DRB200 Digital Reactor Block and spectrophotometer UV–Vis (Thermo Fisher Scientific, UV23000)
TSS	Standard method 2540D	Glass fiber filter
TDS	Electrometric	Water quality meter (AZ 86031)
Color	Pt/Co method	Spectrophotometer UV–Vis (Thermo fisher scientific, UV23000)
Conductivity	Electrometric	Water quality meter (AZ 86031)

**Table 2 membranes-11-00066-t002:** Membrane composition.

Membrane	Composition		
	PSf (g)	PVP (g)	NMP (mL)
Pristine PSf	7.5	0	42.5
PSf/PVP 0.15	7.35	0.15	42.5
PSf/PVP 0.25	7.25	0.25	42.5
PSf/PVP 0.35	7.15	0.35	42.5

**Table 3 membranes-11-00066-t003:** Comparison of the results of measurements of membrane thickness, porosity, average pore size, and water contact angle (WCA).

Membrane	Membrane Thickness (L, µm)		Porosity (ε, %)		Average Pore Size (r_m_, nm)	WCA (°)
	SEM	Dial Thickness	SEM	Dial Thickness		
PSf	116 ± 9.0	158 ± 0.03	30.52 ± 2.4	24.44 ± 3.1	5	120 ± 0.5
PSf/PVP 0.15	107.5 ± 5.5	150 ± 0.01	44.77 ± 4.4	32.80 ± 2.1	17	65.37 ± 1.2
PSf/PVP 0.25	101 ± 2.0	106 ± 0.05	71.94 ± 9.8	67.88 ± 3.4	13	50.46 ± 0.8
PSf/PVP 0.35	148 ± 5.0	226 ± 0.01	39.69 ± 1.34	26.19 ± 1.3	12	50.26 ± 3.4

**Table 4 membranes-11-00066-t004:** The results of mechanical strength test of the membranes.

Membrane	Tensile Strength (MPa)	Elongation at Break (%)
PSf	5.41 ± 0.96	33.47 ± 9.75
PSf/PVP 0.15	4.63 ± 0.45	36.14 ± 12.39
PSf/PVP 0.25	4.18 ± 0.45	22.90 ± 14.90
PSf/PVP 0.35	3.89 ± 0.44	27.34 ± 7.34

**Table 5 membranes-11-00066-t005:** Comparison of the batik wastewater characteristics.

Parameter	Unit	Initial Wastewater	After Coagulation–Flocculation	pH 7 Adjustment	After Conventional Filtration
COD	mg/L	5100	1100	1215	674
TDS	mg/L	2200	2250	2670	2590
TSS	mg/L	700	325	362	3
Conductivity	mS	4.20	4.39	5.34	5.18
Color	mg/L	8500	378	405	405
pH	–	9.40	4.09	6.99	7.00

## Data Availability

The data presented in this study are available on request from the corresponding author. The data are not publicy available due to restrictions from the batik industry.

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
