# Peer review of "Polysulfone–Polyvinyl Pyrrolidone Blend Polymer Composite Membranes for Batik Industrial Wastewater Treatment"

_membranes, 2021, doi:10.3390/membranes11010066_

Round 1

Reviewer 1 Report

The manuscript "Polysulfone-Polyvinyl pyrrolidone Blend Polymer Composite Membranes for Batik Industrial Wastewater Treatment"  should be seriously revised considering the following aspects:

  1. The abstract should be revised so as to make clear if the membranes and experiments have been undertaken at laboratory scale and which is the main objective of this study and its results.
  2. The terminology should be used consistently and in accordance with the well known and accepted international terminology. Few examples in this respect are: a) the term "rejection of BOD, COD etc" should be replaced with "removal efficiency" the adequate term used for wastewater treatment processes. This correction should be also used in Eq.5 where it is called "Batik wastewater removal" instead of removal efficiency; b) the same stands for "reversible and irreversible resistance of the membrane" that should be replaced with "reversible and irreversible membrane fouling".
  3. The references that support the Introductory chapter are quite old, although there is a lot of international scientific literature available in the last 5 years (including reviews) for membrane processes and their applications for textile wastewater treatment.
  4. Introduction should be revised and better structured to present the background information for this study, i.e. types of membrane processes adequate for textile wastewater treatment, main problems, process combinations, etc. 
  5. There are neither clear objectives nor the novelty stated for this study (the novelty should be discussed considering other international research in the same field) and these aspects should be clarified at the end of the Introduction section. Now, this study presents so many information, but lacks a clear background and structure that would facilitate its understanding.
  6. The Materials and Methods section is not clear. It lacks: a) a table with all the water quality indicators, equipment and methods of analysis, b) information about how the coagulation dose was calculated, c) what type of membrane was obtained, d) How was the normal filtration has been carried out and on which media ?, e) what kind of conditions were used for membrane filtration and for backwashing?, etc.
  7. English should be seriously revised and the support of a native speaker is highly recommended. There are many sentences where the meaning is not clear due to bad English.

Reviewer 2 Report

This paper introduces a preparation method of PSf/PVP composite membrane, which is used to treat practical batik wastewater. The overall idea is clear and the experimental scheme is relatively complete. But the article lacks obvious innovation, and there are some questions as follows.

  1. Page 3, line 95-96: What is the principle of placing the membrane in ethanol solution? What's the difference between 50% and 96% ethanol solution?
  2. What is the PSF / PVP composite membrane prepared in this paper? Microfiltration, ultrafiltration or nanofiltration?
  3. Page 5, Paragraph 3.1: I did not get the result expressed by the author from Fig. 2: ‘the changes in the membrane after the addition of PVP can be seen in the formation of pores in the image with 2500 times magnification. The addition of PVP to the PSf membrane increased the pore structures in the membrane.’It is suggested that the author mark the content you want to express with conspicuous marks in the diagram.
  4. Page 8, Table 2: The authors should explain why WCA decreases rapidly from 65.37 to 50.46°when PVP is added from 0.15 to 0.25, and remains unchanged when it continues to increase to PVP 0.35?
  5. Page 8, Fig. 4: The icon of PSF/PVP 0.15 is given in the infrared image, but the corresponding curve is not drawn. The peak represented by PVP is not reflected in the composite membrane, is it due to too little addition?
  6. Page 9, line 53: What are the reasons for choosing PAC as coagulant? The advantages of the coagulant can be added in the introduction.
  7. Page 10, Fig. 5: It is suggested that the author redraw the drawing. Why is the starting time of red line in Fig. 5(a) different from other lines? Why the medium green line different from other start and end times in Fig. 5(b)? Adjusting the bottom ordinate so that the black line can be seen clearly.
  8. Page 10, Fig. 5: Why is the error bar of PSF / PVP 0.25 water flux curve (red line) significantly higher than the other three lines, and the error bar of each data is the same?
  9. The data in Figure 9, 11 lacks error bars.

Reviewer 3 Report

The present manuscript shows 4 types of membranes prepared and their corresponding properties as RO or such a filtration purpose. Unfortunately, I find several critical points of this study, as below, thus rejection is concluded.

1) PVP is water soluble. The authors seem not to be considering of this effect, leading to a collapse of the research story.

2) Prepared membranes are very immature. Please take care of repeatability for membrane preparation. Membrane characterization is the next.

As a hint, Fig.4, which shows independence of PVP intensity as a function of PVP addition, is a typical result to prove almost all PVP are rinsed out of PSf. The manuscript should report based on PSf pore film by way of removal of PVP from PSf and PVP blend membranes, through phase inversion method.

Round 2

Reviewer 1 Report

The manuscript was improved based on the Reviewers' suggestions. In its revised form, I consider that this study may be published as it is.

Author Response

Thank you very much for the valuable comments.

Reviewer 2 Report

The author seriously repThe author answers the question of my premise seriously. I agree to accept it.

Author Response

(The authors gave the same response as above.)

Reviewer 3 Report

Dear authors,

Thank you for improvement of the manuscript, but I unfortunately do not believe in the result and conclusion, including the methodology and reliability, with the same reason as my previous comments.

According to FTIR, almost all membranes may contain little PVP, the thickness and pore tendency reported is not consistency, the membrane properties are not scientifically reasonable, casting doubt on the reliability of this manuscript. 

The theoretical porosity and thickness should be calculated by way of the theoretical density, removal of PVP, and such. If PVP is capsuled by Polysulfone, leading to unreaching behavior, please discuss it. The name inclusion of PVP is very questionable, as addition of PVP is a technique, not a resulted polymer membrane.

For menbrane preparation, averagely N=10 is considered, whereas I feel this manuscript looks N=1. Thus, rejection is concluded.

Round 3

Reviewer 3 Report

As the authors claim the repeatability is confirmed, it will be fine to accept the present manuscript for publication, as it is.